Genome-wide analyses of the NAC transcription factor gene family in Acer palmatum provide valuable insights into the natural process of leaf senescence

Meng Xin 1 2 3
Feng Chun 1
Chen Zhu 3
Shah Faheem Afzal 3
Zhao Yue 1 3
Fei Yuzhi 3
Zhao Hongfei 2 hfzhao@zjsru.edu.cn
Ren Jie 3 renjieaaas@sina.com
1 School of Forestry & Landscape Architecture, Anhui Agricultural University , Hefei, Anhui , China
2 College of Urban Construction, Zhejiang Shuren University , Hangzhou, Zhejiang , China
3 Institute of Agricultural Engineering, Anhui Academy of Agricultural Sciences , Hefei, Anhui , China
Thomas Jonathan
Electronic publication date: 2025 Jan 13
Publication date: 2025
Volume: 13
Electronic Location ID: e18817
Received 2024 Aug 6; Accepted 2024 Dec 14
Copyright: © 2025 Meng et al.
Copyright year: 2025
Copyright holder: Meng et al.
License: This is an open access article distributed under the terms of the Creative Commons Attribution License, which permits unrestricted use, distribution, reproduction and adaptation in any medium and for any purpose provided that it is properly attributed. For attribution, the original author(s), title, publication source (PeerJ) and either DOI or URL of the article must be cited.
License URL: https://creativecommons.org/licenses/by/4.0/

Keywords: Acer palmatum, Leaf senescence, Expression profile, NAC gene family, Phylogenetic analysis

Funding: National Natural Science Foundation of China 32301660 and 32271914 This work was supported by the National Natural Science Foundation of China (grant numbers 32301660 and 32271914). The funders had no role in study design, data collection and analysis, decision to publish, or preparation of the manuscript.

==============================
Acer palmatum is a deciduous shrub or small tree. It is a popular ornamental plant because of its beautiful leaves, which change colour in autumn. This study revealed 116 ApNAC genes within the genome of A. palmatum. These genes are unevenly distributed on the 13 chromosomes of A. palmatum. An analysis of the phylogenetic tree of Arabidopsis thaliana NAC family members revealed that ApNAC proteins could be divided into 16 subgroups. A comparison of ApNAC proteins with NAC genes from other species suggested their potential involvement in evolutionary processes. Studies suggest that tandem and segmental duplications may be key drivers of the expansion of the ApNAC gene family. Analysis of the transcriptomic data and qRT‒PCR results revealed significant upregulation of most ApNAC genes during autumn leaf senescence compared with their expression levels in summer leaves. Coexpression network analysis revealed that the expression profiles of 10 ApNAC genes were significantly correlated with those of 200 other genes, most of which are involved in plant senescence processes. In conclusion, this study contributes to elucidating the theoretical foundation of the ApNAC gene family and provides a valuable basis for future investigations into the role of NAC genes in regulating leaf senescence in woody ornamental plants.

Introduction

Acer palmatum is a deciduous tree belonging to the genus Acer in the family Aceraceae (Gao et al., 2020; Tao et al., 2020). It is widely cultivated for its ornamental value, primarily in China, Japan, and Korea (Areces-Berazain, Hinsinger & Strijk, 2021; Contreras & Shearer, 2018). Clinical studies suggest that extracts from A. palmatum, known for their anti-proliferative and antioxidant properties, hold promise for application in cancer treatment or prevention (Bi et al., 2016). In addition to its medicinal applications, the captivating beauty of A. palmatum makes it a popular ornamental plant (Cao, De Craene & Chang, 2020; Xie et al., 2023). The widespread Acer palmatum has attracted significant research attention.

Leaf senescence is an active process that is highly regulated by genes (Ali, Gao & Guo, 2018). Transcription factors (TFs) act as mediators, regulating dramatic shifts in gene expression during the leaf senescence process (Guo, 2013). TFs are proteins that bind to specific cis-regulatory elements in promoter regions or interact with other regulators to activate or inhibit target genes (Manna et al., 2021). The acronym NAC stands for three founding members (NAM, ATAF1/2, and CUC2), which were originally found to contain specific NAC domains (Nakashima et al., 2012). NAC proteins each have a highly conserved N-terminal NAC domain for DNA binding and a variable C-terminal region that determines their activation or repression function (Shao, Wang & Tang, 2015). The NAC domain can be further subdivided into subdomains A–E (Ooka et al., 2003). Research suggests that subdomains C and D may play critical roles in DNA binding, whereas subdomain A may contribute to the composition of homodimers or heterodimers. Subdomains B and E likely contribute to the functional diversity within NAC proteins (Olsen et al., 2005).

The NAC protein, which is unique to plants, is essential for the growth and development of plants and represents one of the largest families of TFs among plant regulators (Singh et al., 2021). NACs have emerged as crucial regulators of leaf senescence across various plant species (Guo & Gan, 2006; Mao et al., 2017). Research suggests that NAC TFs promote age-dependent senescence by activating genes associated with this process (Nagahage et al., 2020; Wang et al., 2021). Additionally, NAC proteins influence senescence by regulating genes involved in the production of gibberellin (GA), a hormone that impacts chlorophyll degradation (Fan et al., 2020, 2021). Recent studies have further elucidated the role of NAC TFs in dark-induced senescence, wherein they participate by influencing chlorophyll breakdown (Chou et al., 2018; Sun et al., 2024).

The leaves of A. palmatum exhibited a vibrant green hue during the summer months. However, as the autumn season approaches and temperatures begin to decrease, these leaves undergo a transformation, turning a striking shade of red before falling to the ground (Xie et al., 2023). Leaf senescence, the final step of in a leaf’s life cycle, is essential for the plant life cycle. Although the crucial role of NAC family transcription factors in leaf senescence has been widely illustrated in many plant species, it has not been studied in A. palmatum. In this study, we identified the NAC transcription factor gene family in A. palmatum and analysed the cis-regulatory elements, phylogenetic relationships, gene structures, and chromosomal locations of these NAC genes. Furthermore, we investigated the expression patterns of ApNACs in both summer (nonsenescent) and autumn (senescing) leaves of A. palmatum using transcriptomic data and revealed putative candidate ApNAC genes that regulate leaf senescence. This research might elucidate the molecular mechanisms underlying ApNAC gene regulation of leaf senescence in A. palmatum and provide a foundation for further exploration of the broader functions of NAC genes.

Materials and Methods

Plant material and treatment

A. palmatum seedlings cultivated in the experimental field of the Anhui Academy of Agricultural Sciences, Hefei city, Anhui Province, China, were used as study samples (31.86°N, 1117.27°E). We collected green leaves (mature leaves) and red leaves (senescing leaves) in the summer and autumn. Three biological replicates of each sample (fiive leaves of A. palmatum) were collected. The samples were immediately transferred to liquid nitrogen and then kept at −80 °C for future utilization in qPCR analysis.

Database search and sequence retrieval

The A. palmatum genome data were obtained from the A. palmatum genome database (accession number: PRJNA850663; https://www.ncbi.nlm.nih.gov/). The Pfam database (http://pfam.xfam.org/) was searched for HMM maps of the NAC domain (PF02365). The conserved domains of the candidate gene sequences were verified using the CD-Search tool (https://www.ncbi.nlm.nih.gov/cdd/) and BLASTP searches. ExPASy (https://www.expasy.org) was used for analysis to predict the physicochemical characteristics of the ApNAC proteins. Subcellular localization was ascertained using the CELLO v2.5 website (http://cello.life.nctu.edu.tw/; Yu et al., 2006).

Phylogenetic analysis of ApNAC proteins

The NAC protein sequences for Arabidopsis thaliana were downloaded from the TAIR database (https://www.arabidopsis.org/) for comparative evolutionary analysis. Protein sequences from A. palmatum and A. thaliana were used to generate an unrooted phylogenetic tree via the neighbour-joining (NJ) method and 1,000 iterations of bootstrap testing. This analysis allowed us to classify all ApNAC proteins into distinct subgroups based on the established Arabidopsis NAC protein classification system.

Conserved motifs and gene structure analysis

We utilized the Multiple EM for Motif Elicitation (MEME) service (http://meme-suite.org/tools/meme) to search for conserved motifs in NAC proteins. To understand the intron‒exon organization of each ApNAC gene, their genomic sequences were aligned with the corresponding gene structure annotation files. TBtools software facilitated the visualization of both the conserved motifs and the intron‒exon structures.

Chromosomal location, duplication events, and homology

Genome annotation files revealed the chromosomal positions of the ApNAC gene family, which were then visualized using the MG2C website (http://mg2c.iask.in/mg2c_v2.1/). We examined the genomic sequences and GFF files for A. thaliana, Populus tomentosa, Citrus sinensis, and Vitis vinifera, which we obtained from the NCBI database (https://www.ncbi.nlm.nih.gov/). ApNAC gene duplication events were analysed using the MCScanX program (Wang et al., 2012). ApNAC gene homology relationships were examined with Dual Synteny Plotter software (Chen et al., 2020).

Analysis of promoter cis-regulatory elements

To elucidate the potential regulatory mechanisms governing ApNAC gene expression, we investigated 2 kilobase (kb) regions upstream of the transcription start sites for the ApNAC gene. The PlantCARE database (http://www.plantcare.co.uk/) was used to identify potential cis-acting regulatory elements. This analysis aimed to predict the presence of crucial cis-regulatory elements involved in influencing plant development and stress and hormone responses. TBtools software was subsequently used to visualize the 22 most common cis-regulatory elements found within the ApNAC gene promoters (Chen et al., 2020).

Analysis of ApNAC gene expression patterns

Transcriptomic data were downloaded from the NCBI database (accession number: PRJNA850663). The relative gene expression values were expressed as fragments per kilobase million (FPKM) values, and all the transcriptomic data were converted to log2(FPKM+1) values. For ApNAC genes that were differentially expressed in response to senescence stress, the selection criterion was a log2-fold change in value. The TBtools program was used to generate a heatmap of the transcript profiles of the ApNAC genes (Chen et al., 2020).

QRT–PCR validation of ApNAC gene expression levels

Total RNA was isolated from the summer and autumn leaves of A. palmatum using a commercially available plant RNA extraction kit (ZD02001; Zoonbio, Nanjing, China). The quality and purity of the RNA were confirmed using a NanoDrop spectrophotometer (Thermo Fisher Scientific, Waltham, MA, USA). A Hifair® First Strand cDNA Synthesis Kit (11123ES10; Yeasen, Shanghai, China) was used to generate the corresponding cDNA. Then, qRT‒PCR was performed using SYBR Premix Ex Taq (RR420A; Tli RNaseH Plus, TaKaRa, Dalian, China) on a Lightcycle K Real-Time PCR Detection System (BIOER, Hangzhou, China). The reactions were performed as follows: 40 cycles of 95 °C for 5 s, 60 °C for 30 s, and 72 °C for 20 s. Primer Premier 5 software was used to design gene-specific primers (Table S1; Lalitha, 2000). Actin served as an internal control gene. Three biological replicates were designed for each sample (five leaves of A. palmatum). The gene expression level was determined using the 2−∆∆Ct method developed by Livak & Schmittgen (2001) (Table S2).

ApNAC gene correlation analysis

To explore the various TFs that participate in leaf senescence associated with ApNAC genes, we employed R to calculate transcriptome correlations via Pearson correlation coefficient (PCC) analysis. The expression profiles of the ApNAC and non-ApNAC genes were compared in this manner. This analysis aimed to identify genes whose expression patterns closely mirrored those of the ApNAC genes. We subsequently employed Cytoscape software (version 3.6.1) for coexpression network visualization (Shannon et al., 2003).

Results

Identification and characterization of ApNAC genes

Using the hidden Markov model (HMM) profiles of the NAC (PF02365) domain as queries to scan the A. palmatum genome, 116 potential ApNAC genes were found in the genome of A. palmatum. Based on their chromosomal positions, these genes were designated ApNAC1 through ApNAC116 (Table S3). We examined the lengths, MWs and pIs of the encoded proteins and utilized an online platform for subcellular localization prediction. The smallest ApNAC protein (ApNAC28) harboured only 139 amino acids with a corresponding MW of 15,833.98 kDa, whereas the largest protein (ApNAC108) contained 871 amino acids and had a predicted MW of 98,111.96 kDa. The pI values of these ApNAC proteins exhibited a broad range, from 4.22 (ApNAC114) to 10.02 (ApNAC50). Subcellular localization predictions revealed diverse localizations of the ApNAC proteins: ApNAC77 was predicted to reside in the extracellular space; four proteins (ApNAC42, ApNAC50, ApNAC114, and ApNAC115) were predicted to localize to chloroplasts; and 15 proteins (ApNAC32, ApNAC33, ApNAC34, ApNAC38, ApNAC40, ApNAC48, ApNAC56, ApNAC57, ApNAC60, ApNAC61, ApNAC64, ApNAC97, ApNAC98, ApNAC99, and ApNAC116) were predicted to reside in the cytoplasm, with the remaining ApNAC proteins predicted to be localized within the nucleus (Table S3).

Classification and phylogenetic analysis of ApNAC proteins

To look at the evolutionary connections between ApNAC family members and other recognized plant NAC proteins, we generated a phylogenetic tree using sequence alignments of 116 ApNAC and 105 Arabidopsis ANAC proteins (Fig. 1). Our analysis grouped the ApNAC proteins into 16 distinct subfamilies based on their homology with Arabidopsis NAC proteins. These subfamilies included ATAF, ANAC3, NAP, ONAC022, NAM, NAC1, OsNAC7, ANAC001, TIP, ANAC011, NAC2, ONAC003, ANAC063, SENU5, TERN, and OSNAC8 (Fig. 1). The ANAC063 subfamily had the greatest number of ApNAC members (20), whereas the NAC1 subfamily contained only one. This diversity within the ApNAC family mirrored observations previously reported in A. thaliana (Ooka et al., 2003).

Figure 1 Phylogenetic tree of NAC proteins in A. palmatum and Arabidopsis.

Based on the protein sequences of A. palmatum and Arabidopsis, a neighbor-joining (NJ) phylogenetic tree was constructed using the MEGA7.0 program. A total of 16 subfamilies were created from the tree, and each subfamily was given a unique colour and label.

Gene structure and motifs analysis of ApNAC proteins

By revealing the functional domains of the ApNAC proteins, we identified 20 conserved motifs using the MEME program, which were designated motifs 1–20 (Fig. 2A). All the motifs resided within the well-conserved N-terminal NAC domain. Motif 7 was universally present across all ApNAC protein families, whereas motifs 1 and 8 were prevalent among most members. The number of motifs per protein varied, with ApNAC34 containing the fewest (one motif) and ApNAC107 harbouring the most (10 motifs). Most of the closely related members of the phylogenetic tree presented similar motif compositions. For example, members of the AtNAC3, SENU5, and OsNAC8 subfamilies all possessed motif 7, suggesting that members grouped in the same clade based on similar conserved motifs might share similar functions.

Figure 2 Phylogenetic tree, conserved domains, motifs, and intron-exon gene structure of the A. palmatum NAC genes.

(A) The MEME motifs were shown as different coloured modules at the N-terminal indicating the NAC domain region. (B) Conserved domains of ApNAC proteins. (C) Exons are indicated by green boxes, and introns are indicated by black lines.

To explore structural diversity within the ApNAC gene family, we analysed intron‒exon distribution patterns (Fig. 2C). Subfamilies generally exhibited concordance in intron‒exon structures and gene length. The number of introns varied between 0 and 11, with two genes lacking introns entirely and 20 genes containing just one intron. Genes with two introns were the most abundant, whereas 10 genes (ApNAC36, ApNAC65, ApNAC69, ApNAC56, ApNAC57, ApNAC34, ApNAC78, ApNAC76, ApNAC77, and ApNAC79) presented more than six introns.

Chromosomal localization, repeat events, and synteny analysis of ApNAC genes

To visualize the chromosomal locations of the ApNAC genes, we generated a distribution map using MG2C software (Fig. 3). The 116 ApNACs displayed an uneven distribution across the 13 chromosomes, with no apparent correlation between the number of genes on each chromosome and its length. Chromosome 3 contained the greatest number of ApNAC genes, with 18, followed by chromosome 12, with 13 ApNAC genes, whereas chromosome 1 had the fewest, with three ApNAC genes.

Figure 3 Chromosomal locations of ApNAC transcription factors.

A total of 116 ApNAC genes are distributed throughout 13 chromosomes. The chromosomes of A. palmatum are represented by bars. The chromosome number is located above each chromosome. The chromosome length is indicated by the scale on the left.

To determine repeat events in ApNAC genes, we performed homology analysis using MCScanX software. Segmental duplications were identified among the 116 ApNAC members. Chromosome 5 presented the highest abundance of segmental duplication gene pairs (3), followed by chromosomes 4 and 2 (two pairs each) (Fig. 4 and Table S4). Additionally, we identified 21 pairs of tandem duplications located on chromosomes 2, 3, 4, 6, 7, 9, 11, and 13 (Fig. 3 and Table S5). These findings suggest that segmental and tandem duplications are likely the primary causes driving ApNAC gene family expansion.

Figure 4 Chromosomal relationships of ApNAC genes shown schematic.

Pairs of ApNAC genes with segmental duplications are shown by coloured lines. Gene density information is represented by the heatmap and line graph, where red denotes high gene density and yellow denotes low gene density.

To gain deeper insights into ApNAC gene evolution, we constructed a comparative homology map using NAC genes from A. thaliana, P. trichocarpa, C. sinensis, and V. vinifera. This analysis revealed collinearity between ApNAC and NAC genes from these plant species: P. trichocarpa (134 homologous gene pairs), A. thaliana (69 pairs), C. sinensis (90 pairs), and V. vinifera (81 pairs) (Fig. 5).

Figure 5 NAC genes synteny analyses between four representative plants and A. palmatum.

Gray lines in the background represent collinear blocks between the genomes of A. palmatum and other plant species, while red lines represent pairs of NAC genes with segmental duplications.

Cis-acting regulatory element (CARE) analysis of ApNAC genes

To investigate the possible cis-regulatory elements involved in ApNAC gene expression, we extracted 2 kilobase sequences upstream of the promoter regions for all ApNAC genes. Putative cis-regulatory elements within the promoter region sequences were identified using the PlantCARE database (Fig. 6 and Table S6). This analysis revealed a diverse array of cis-regulatory elements associated with various biological processes, including developmental stages, plant hormone signalling, and abiotic stress responses. The cis-regulatory elements associated with plant development were predominantly responsive to light and specific to meristematic tissues. Additionally, cis-regulatory elements responsive to hormones such as ethylene, abscisic acid (ABA), jasmonic acid (JA), and salicylic acid (SA) were identified. These findings suggest that ApNAC genes, potentially under hormonal regulation, might be essential for the successful growth, development, and stress tolerance of A. palmatum.

Figure 6 Analysis of ApNAC gene promoter cis-elements statistically.

The colour and number of the grid indicate the number of cis-elements in the promoter region of ApNAC gene.

Expression patterns of ApNAC genes during leaf senescence

To explore the potential roles of ApNACs in regulating leaf senescence, we analysed their expression patterns using transcriptome analysis and qRT‒PCR method. The transcriptomic analysis revealed that the expression levels of multiple genes, such as ApNAC02, ApNAC04, ApNAC05, ApNAC06, ApNAC41, ApNAC48, ApNAC51, ApNAC83, ApNAC91, and ApNAC100 were significantly increased in senescing leaves (autumn) (Fig. 7 and Table S7).

Figure 7 The ApNAC genes implicated in autumn leaf senescence are represented in a hierarchical clustering heatmap.

Analysing the RNA-Seq data resulted in the creation of a heatmap on the basis of the log2 fold change values in summer and autumn. A colour gradient in the upper right corner depicts the shift in expression levels from blue (downregulated) to red (upregulated).

We confirmed the expression levels of the top 10 highly expressed ApNACs through qRT-PCR analysis. Compared with those nonsenescing leaves (summer), the relative expression levels of ApNAC02, ApNAC04, ApNAC05, ApNAC06, ApNAC41, ApNAC48, ApNAC51, ApNAC83, ApNAC91, and ApNAC100 were increased in senescing leaves (autumn) (Fig. 8A). Therefore, different ApNACs have significant effects on the leaf senescence of A. palmatum.

Figure 8 The expression profiles of fifteen representative ApNAC genes were analysed using qRT‒PCR.

(A) The relative expression levels of 10 ApNACs were increased in senescing leaves (autumn). (B) The relative expression levels of five ApNACs were decreased in senescing leaves (autumn). The seasons are represented by the x-axis, while the relative expression is shown by the y-axis. The Actin gene was utilized to standardize the qRT‒PCR data. The standard deviation (SD) of the three biological replicates is represented by the bar chart.

Construction of a coexpression network for ApNAC genes

To elucidate the regulatory networks of ApNAC gene function during leaf senescence, we employed R language to analyse the expression levels of the TFs within the A. palmatum leaf transcriptomic data. This analysis aimed to identify coexpressed TFs of ApNAC genes. Two hundred significantly expressed TFs exhibited strong correlations with 10 ApNAC genes (|PCC| > 0.95) (Fig. 9 and Table S8). The top five coexpressed TF families were WRKY (58), AP2/ERF (51), bZIP (41), MYB-related (37), and C2H2 zinc finger (11). These findings suggest that ApNAC genes may interact with these TF families to establish regulatory networks, potentially acting cooperatively to promote leaf senescence.

Figure 9 ApNAC gene coexpression network built with transcriptomic data.

The number of related genes is represented by the hue, with lighter purple denoting fewer related genes and darker purple denoting more related genes.

Discussion

NAC genes act as critical players in fundamental development processes and in the response to environmental stresses (Nuruzzaman, Sharoni & Kikuchi, 2013). NAC genes have been identified in Arabidopsis, rice, and barley (Yuan et al., 2019). However, whether NAC TFs play various roles in regulating the senescence of A. palmatum leaves remains unclear. In the present study, the characteristics of ApNAC genes at the genome level were investigated, and their significant effects on leaf senescence in A. palmatum. Our research findings provide new perspectives for researchers to explore the roles of NAC genes in regulating leaf senescence in woody ornamental plants.

To elucidate the potential functional relationships among the ApNAC genes, we employed the neighbour-joining (NJ) method to generate an unrooted phylogenetic tree based on protein sequences. The results revealed that all 116 ApNAC proteins were grouped into 16 subgroups. It has been reported that genes regulating leaf senescence are phylogenetically clustered together, suggesting that NAC genes with similar biological functions are closely related (Hu et al., 2015; Wei et al., 2016). Two subgroups, ATAF and NAP, contained the most ApNAC members in both A. palmatum and Arabidopsis which have been linked to leaf senescence regulation in other plant species (Garapati et al., 2015; Guo & Gan, 2006), suggesting a potential role for ATAF and NAP subfamily members in A. palmatum leaf senescence. Analysis of RNA sequencing (RNA-Seq) data revealed that ApNAC genes present significant transcriptional responses throughout A. palmatum leaf senescence. To validate these findings, we further investigated the expression profiles of 15 ApNAC genes from various subgroups by qRT‒PCR analysis. The results showed that the relative expression of 10 genes were increased in senescing leave, whereas the relative expression of five genes were decreased.

The increases in ApNAC gene expression levels were caused mostly by tandem and segmental duplication. Tandem duplications have been reported to play significant roles in the expansion of the NAC gene family in Oryza sativa (Nuruzzaman et al., 2010) and Eucalyptus grandis (Hussey et al., 2015). Segmental duplications may also contribute to NAC gene family expansion, as observed in Panicum miliaceum (Shan et al., 2020) and Vigna radiate (Tariq et al., 2022). A total of 116 ApNAC genes were found to have 21 pairs of tandemly duplicated genes and 11 pairs of segmentally duplicated genes. These duplication events were likely driving forces in the diversification and expansion of the NAC gene family within A. palmatum. The A. palmatum genome experienced an ancient whole-genome duplication event (γ-genome-wide replication) (Tao et al., 2020), but evidence of recent independent whole-genome duplication events is lacking (Chen et al., 2023).

NAC TFs constitue one of the largest groups of plant regulatory proteins that can either activate or repress gene expression, affecting how plants react to biotic and abiotic stressors (Nuruzzaman, Sharoni & Kikuchi, 2013). In this study, promoter analysis revealed ABREs within the promoter regions of eight ApNAC genes (ApNAC07, ApNAC22, ApNAC59, ApNAC64, ApNAC72, ApNAC96, ApNAC103, and ApNAC114). In Arabidopsis, ABREs have been linked to ABA-responsive stress signalling (Jensen et al., 2010), and NAC-TFs are known to respond to ABA (Nuruzzaman, Sharoni & Kikuchi, 2013). These findings suggest a potential role for ABA signalling in regulating ApNAC gene expression during leaf senescence in A. palmatum.

RNA-Seq data analysis revealed that the expression levels of ApNAC genes changed significantly during the senescence process in A. palmatum. Among these ApNACs, 10 members were found to be upregulated, whereas four members (ApNAC80, ApNAC116, ApNAC15, and ApNAC79) were downregulated (Fig. 8B), indicating that NACs might play both positive and negative regulatory roles (Xu et al., 2022; Shah et al., 2024). GmNAC06 (from Glycine max) and NaNAC29 (from the wild tobacco Nicotiana attenuata) play positive roles in regulating leaf senescence (Fraga et al., 2021; Ma et al., 2021). Studies have shown that phylogenetic analysis helps predict gene function by identifying similar genes across multiple species (Cao et al., 2023; Yuan et al., 2020). Both GmNAC06 and NaNAC29 belong to the AtNAP subfamily and share homology with ApNAC02, suggesting that ApNaC02 may play a crucial role in regulating leaf senescence in A. palmatum.

Numerous families of TFs, including WRKY, AP2/ERF, bZIP, and NAC TFs, play an essential roles in leaf senescence (Chen et al., 2012; Nakashima et al., 2012; Puranik et al., 2012; Rushton et al., 2012; Licausi, Ohme-Takagi & Perata, 2013; Wang et al., 2018). In A. palmatum, our analysis using the Pearson correlation coefficient (PCC) identified potential coregulators of NAC genes during leaf senescence. These candidates included 58 WRKY, 51 AP2/ERF, 41 bZIP, 37 MYB-related, and 11 C2H2 zinc finger TFs that exhibited strong positive correlations with NAC expression. These findings suggest the existence of a complex regulatory network involving multiple interacting TF families that regulate leaf senescence. For example, OsWRKY5 in rice has been shown to accelerate chlorophyll degradation, promoting leaf senescence in rice (Oryza sativa) by indirectly upregulating the expression of NAC genes related to leaf senescence, such as OsNAP and OsNAC2 (Kim et al., 2019). In Arabidopsis, MYB108 directly regulates genes associated with senescence by binding to the ANAC003 promoter (Chou et al., 2018). The findings presented here underscore the importance of exploring the interplay of TFs in the regulatory mechanisms of A. palmatum leaf senescence.

Conclusions

In A. palmatum, 116 NAC genes were identified in this study. Using an Arabidopsis NAC protein-based phylogenetic tree, we classified these TFs into 16 distinct subfamilies. Analysis of gene distribution across chromosomes and sequence homology suggested that segmental and tandem duplications were the major contributors to ApNAC gene family expansion. Analysis of the RNA-Seq data revealed distinct expression profiles for the NAC genes. Most ApNAC genes were significantly upregulated during leaf senescence. These ApNACs with prominent expression changes are promising candidates for further investigations into their regulatory roles in A. palmatum leaf senescence. Overall, this research provides a foundation for understanding of the regulatory roles of ApNAC genes, offering valuable insights for future investigations into their biological functions in A. palmatum.

Supplemental Information

Supplemental Information 1 Primer used for qRT-PCR analysis of ApNACs.

Supplemental Information 2 qRT-PCR quantitative data.

Supplemental Information 3 Annotation of A. palmatum NAC transcription factors.

Supplemental Information 4 Details of gene segmental-duplication of ApNACs.

Supplemental Information 5 Details of gene tandem-duplication of ApNACs.

Supplemental Information 6 Cis-acting regulatory element (CARE) analysis of ApNACs.

Supplemental Information 7 Expression patterns of 68 ApNACs during A. palmatum leaf senescense.

Supplemental Information 8 Coexpression network analysis between 10 ApNACs and other genes.

Supplemental Information 9 MIQE Checklist.

Additional Information and Declarations

Competing Interests

Author Contributions

Data Availability

The authors declare that they have no conflict of interests.

Xin Meng performed the experiments, analyzed the data, prepared figures and/or tables, authored or reviewed drafts of the article, and approved the final draft.

Chun Feng conceived and designed the experiments, analyzed the data, prepared figures and/or tables, authored or reviewed drafts of the article, and approved the final draft.

Zhu Chen analyzed the data, authored or reviewed drafts of the article, and approved the final draft.

Faheem Afzal Shah analyzed the data, authored or reviewed drafts of the article, and approved the final draft.

Yue Zhao analyzed the data, authored or reviewed drafts of the article, and approved the final draft.

Yuzhi Fei analyzed the data, authored or reviewed drafts of the article, and approved the final draft.

Hongfei Zhao conceived and designed the experiments, authored or reviewed drafts of the article, and approved the final draft.

Jie Ren conceived and designed the experiments, authored or reviewed drafts of the article, and approved the final draft.

The following information was supplied regarding data availability:

The raw data is available in the Supplemental Files.

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
