# Peer review of "Genome-wide analyses of the NAC transcription factor gene family in Acer palmatum provide valuable insights into the natural process of leaf senescence"

_PeerJ, doi:10.7717/peerj.18817_

## Round 0.1 · original submission · Major Revisions

While all three reviewers agreed that the topic of the manuscript was of interest, each had issues with various part of the manuscript.

Please address all their comments

Reviewer 1 ·

Basic reporting

the manuscript is basically in order and the topic covered in the manuscript is quite timely and interesting. The text is clear and the illustrations are in order.
However, the references to the literature could be expanded, because, for example, there are hardly any data on the literature results of the last 5 years. Most of the literature is more than 10 years old. It would also be useful to update this in the Introduction and Discussion chapters.
The hypotheses are in order and so are the conclusions.

Experimental design

The aim of the research is defined and the materials and methods are appropriate. It would be worthwhile to include one or two diagrams of the trees studied and their locations with GPS coordinates in the manuscript.

Validity of the findings

The findings are correct, well written and clear. The conclusions are also sound. The Discussion chapter would be worth expanding, as I indicated above.

Additional comments

no comments

·

Basic reporting

The study entitled “Genome-wide analyses of the NAC transcription factor gene family in Acer palmatum provide valuable insights into the natural process of leaf senescence’’ investigate possible roles of NAC TF in regulating leaf senescence. The idea, methodology and flawas of novelty may look fresh. But, there are significant issues with terminology, writing, and language (poor English).

Introduction:
1.The semantic integrity in the introduction is very weak and the transitions between paragraphs are very sharp.

2. The hypothesis of the study should be stated more clearly.

Experimental design

Generally experimental design look good, but there are some minor issues to be adressed as stated below:

1. Please check the sentence and clearly state the number of replicate and the number of examples in each replicate.

2. please write cDNA shynthesis and qPCR steps respectively.

4. 2-∆∆CT? (line 169).

Validity of the findings

1. It will be more beneficial if tables and figures are given in the results section, where possible, so that the reader could follow the results more easily.

2. It is a shortcoming that the study did not mention any downregulated genes and their possible functions.

3. how were these genes selected? (line 262).

4. Perhaps the most important finding of the study is that ApNAC103 and ApNAC62 may be negative regulators. However, heatmap-based relative expression results show that these genes are upregulated in autumn leaves. Please mention the possible reasons for this situation.

5. I strongly recommend you to perform qRT-PCR reaction for the validation of expression profile of negative regulator, at least for some of them.

6. In the discussion section, it seems that the results of the study are insufficiently compared with the current literature. In this context, please use more current literature and expand the discussion section.

Additional comments

1. The results mentioned in the previous sentence seem to belong to this reference. Please remove this reference or check your sentence (line 290).

2. It is not fully understood what is meant (line 281)

3. please check the sentence (line 163).

4. please make full sentence (line 290).

5. please make full sentence (line 322).

6. Please list the references alphabetically and rearrange them according to the journal's rules.

7. Please check all text-in citation according to journals rules.

·

Basic reporting

see the comments on the part of Additional comments

Experimental design

see the comments on the part of Additional comments

Validity of the findings

see the comments on the part of Additional comments

Additional comments

Reviewer's Comments
In this study, the NAC genes from A. palmatum whole genome were identified. Through RNA-Seq data and qRT-PCR, the expressions of the ApNAC genes in A. palmatum were analysed, and identified potential candidates that might regulate leaf senescence. These findings provide a valuable foundation for future genetic engineering efforts of woody ornamental plants. However, there are still some issues that need to be addressed and corrected before the article is published. Here are some suggestions for revisions to this paper:
1. The sentences in lines 52 (The genus Acer, which belongs to the Aceraceae family, is rich in deciduous and evergeen trees and shrubs and is distributed across Asia, Europe, and North America. China serves as the global centre for modern Acer diversity, with over 100 recognized species.) and the sentences in lines 55 (The diverse genus Acer (Aceraceae, Sapindales) includes the species Acer palmatum, cultivated primarily in China, Japan, and South Korea) are semantically redundant, please revise them.
2. The sentence in line 75 (The NAC proteins, which is unique to plants, is essential for the growth and development of plants and represents, one of the largest families of TFs among plant regulators) has inconsistent singular and plural forms, please check and revise it.
3. The title on line 155 (qRT-PCR validation of ApNAC Gene Expression Levels ) should have the first letter of each word capitalized, please make the necessary changes.
4. Regarding the sentence on line 185 “We also examined the lengths, MWs, pIs, and subcellular locations of the encoded proteins.”, the article did not conduct subcellular localization experiments, it only used an online platform for subcellular localization prediction. The description must be accurate.
5. The notation for figures in lines 199 (Fig. 1) and 203 (Figure 1) is inconsistent. Author should maintain a consistent method of representation.
6. The description in the sentence on line 228 (Chromosome 3 harboured the most ApNAC genes (18), followed by chromosome 12 (13), whereas chromosome 1 contained the fewest (3).) is not precise enough. It should clearly state what the numbers in parentheses refer to.
7. The abbreviation in parentheses in the sentence on line 258 (The abbreviation in parentheses in the sentence on line 258 is incorrect; the author needs to carefully check and revise it.) is incorrect. Author needs to carefully check and revise it.
8. The expression in the sentence on line 259 (Especially ApNAC02, 04, 05, 06, 41, 48, 51, 83, 91, and 100, increased during autumn senescence.) is incorrect; It should state that the expression level of the gene increases, not that the gene itself increases.
9. Regarding the sentence on line 262 (To further explore the potential regulators of leaf senescence in A. palmatum, we selected 10 ApNAC genes and analysed their relative transcript abundance via qRT-PCR.), why were these 10 genes chosen, and what is the basis for this selection?
10. The sentence on line 281 (This study we delved into the characterization of ApNAC genes in the A. palmatum genome.) contains a grammatical error, please make the necessary correction.
11. The sentence on line 322 (Previous studies on functional prediction of TFs family members on the basis of phylogenetic analysis) contains a grammatical error, please make the necessary correction.
12. The sentence on line 328 (Compared with senescence-promoting NAC genes, senescence-suppressing NAC genes may present a more intricate evolutionary history.), is the conclusion drawn reasonable?
13. The sentence in the caption of Figure 1 (MEGA 7.0 was used for multiple sequence alignment, and 1000 bootstrap repetitions, pairwise deletion, and p-distance were used to build the phylogenetic tree.) would be more appropriately placed in the Methods section.
14. The sentence in the caption of Figure 3 (Each chromosome has its number on the left side.) does not match the figure.
15. The notation for chromosomes in Figures 3, 4, and 5 is inconsistent, which can lead to misunderstandings; therefore, a consistent method of writing should be used.

---

## Round 0.2 · accepted · Accept

The reviewer has indicated that they are satisfied that all of the reviewers' comments have been addressed. This manuscript is now ready for publication.

Reviewer 1 ·

Basic reporting

The authors have revised the manuscript accordingly. The English is correct and the text is coherent. The hypothesis and the presentation of its results are adequate. The figures are also correct.

Experimental design

It is absolutely fine.

Validity of the findings

The hypothesis is well formulated, and the conclusions are well in line.